# Deep Clustering Activation Maps for Emphysema Subtyping

**Weiyi Xie**                                                                 WEIYI.XIE@RADBOUDUMC.NL
**Colin Jacobs**                                                        COLIN.JACOBS@RADBOUDUMC.NL
**Bram van Ginneken**                                        BRAM.VANGINNEKEN@RADBOUDUMC.NL
*Diagnostic Image Analysis Group, Radboud University Medical Center*

## Abstract

We propose a deep learning clustering method that exploits dense features from a segmentation network for emphysema subtyping from computed tomography (CT) scans. Using dense features enables high-resolution visualization of image regions corresponding to the cluster assignment via dense clustering activation maps (dCAMs). This approach provides model interpretability. We evaluated clustering results on 500 subjects from the COPDGene study, where radiologists manually annotated emphysema sub-types according to their visual CT assessment. We achieved a 43% unsupervised clustering accuracy, outperforming our baseline at 41% and yielding results comparable to supervised classification at 45%. The proposed method also offers a better cluster formation than the baseline, achieving 0.54 in silhouette coefficient and 0.55 in David-Bouldin scores.

**Keywords:** Deep Clustering, Emphysema Clustering, Class Activation Maps.

## 1. Introduction

The Fleischner Society has proposed an emphysema sub-typing system (Lynch et al., 2015) with six subgroups of emphysema. These subtypes can improve understanding of disease heterogeneity. The system categorized parenchymal emphysema into absent, trace, mild, moderate, confluent, or advanced destructive based on visual CT assessment. However, such a system largely depended on prior knowledge of emphysema CT characteristics and this may not fully capture the disease heterogeneity. This study uses a deep learning-based clustering method to find emphysema subtypes in a completely data-driven fashion. Furthermore, we address the issue of the lack of model interpretability in deep learning clustering methods. Unlike existing deep learning clustering methods that operate on high-level semantic features at a reduced resolution from a convolutional classification network, our method exploits dense features from a segmentation network. With dense features we can generate high-resolution class activation maps (Zhou et al., 2016) to visualize image regions reflecting the cluster assignment as a way to interpret the model's decisions.

## 2. Method

We followed DeepCluster (Caron et al., 2018) to design our clustering method. Denote the feature extraction network as $f_\theta$ and a training set $X = x_1, x_2, ..., x_N$ of $N$ training CT scans. All network parameters were randomly initialized before training. For each epoch, We first extracted features for all $x_n$ by forward-passing $x_n$ to $f_\theta$ with frozen parameters. These features were later scaled and PCA-reduced.

Next, k-means clustering was applied based on the extracted features to assign clusters for training examples. The cluster assignment was used as the classification pseudolabel for training a classification head (its parameters are denoted as $g_W$) together with the feature extraction network $f_\theta$ which is now trainable. The parameters in $g_W$ were reset to a random initialization at each epoch before training $g_W$. This adapts to the possible reassignment of k-means across epochs. The training objective is to minimize the multinomial logistic loss $\min_{\theta W} \frac{1}{N} \sum_{n=1}^{N} l(g_W(f_\theta(x_n)), y_n)$, where $y_n$ is the classification pseudo label for $x_n$.

Because classification performance using randomly initialized convolution features is already far above the chance level, starting from this weak signal, DeepCluster bootstrapped the discriminative power of features obtained by $f_\theta$ progressively during training. The feature network $f_\theta$ is a 3D-UNet (Çiçek et al., 2016), which has three down-sampling and up-sampling layers. Each layer in the down-sampling path consists of two convolutions and a max-pooling operation. Following the down-sampling layers, two more convolutions are used to double the number of convolution filters. Then three upsampling layers were applied. Each layer contains one tri-linear interpolation in the up-sampling path, followed by two convolutions to reduce the interpolation artifacts. Convolution filters have $3 \times 3 \times 3$ kernel size, a stride of 1, and zero-padding. Dense features are features after the last up-sampling layer, as the output of $f_\theta$. $g_W$ is a $1 \times 1 \times 1$ convolution with a bias to squeeze dense features to have six channels corresponding to six emphysema subtypes.

We used manually-annotated lung masks to average-pool dense features within the lungs before feeding them into $g_W$ for classification training. We generated dense clustering activation maps by applying $g_W$ on dense features, skipping the lung-wise pooling. The baseline method used a 3D-UNet without up-sampling layers as $f_\theta$. We compensated for the reduced number of parameters in the baseline by adding one more down-sampling layer and more filters. We ran experiments on an NVIDIA A100 with 40 GB GPU memory, using Pytorch library 1.7.1.

## 3. Results

We performed experiments with 2500 subjects from the COPDGene cohort using the baseline inspiration CT scan of each subject. We trained our method using 2000 scans and evaluated on 500 scans. Our test set was a subset of an existing study (Humphries et al., 2020), which reported an accuracy of 45% in classifying the same set of emphysema subtypes trained on manual labels. Table 3 shows that the proposed model reached 43% unsupervised clustering accuracy, outperforming the baseline, and performing comparable to supervised classification accuracy. In terms of internal cluster measurement, we achieved 0.55 in David-Bouldin (the lower, the better) index and 0.54 in silhouette coefficient, showing a substantial improvement over the baseline.

Fig. 3 shows dense clustering activation maps for two different clusters, representing upper-lobe dominant emphysema and sub-pleural paraseptal emphysema.

## 4. Discussion and Conclusion

This paper presented a novel deep learning based clustering method that exploited dense features for clustering. This offers a high-resolution visualization of cluster assignments

| Method | clustering accuracy | silhouette coefficient | Davies-Bouldin score |
|---|---|---|---|
| baseline | 41% | 0.44 | 0.69 |
| proposed | **43%** | **0.54** | **0.55** |

Table 1: Clustering Performance, the best entry shown in bold.

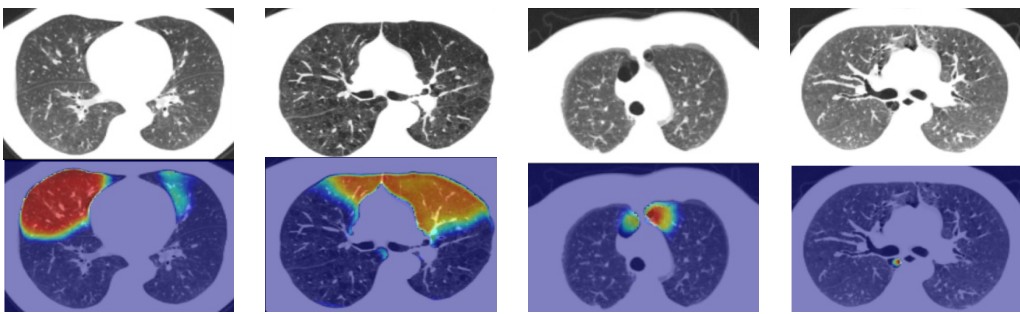

Figure 1: Dense clustering activation maps of four cases from two different clusters. Left two cases: upper-lobe dominance emphysema, the right two: subpleural paraseptal emphysema.

using dense clustering activation maps. The method improved the clustering performance compared with the baseline method. Dense features learned by clustering can be transferred to many possible downstream tasks, not limited to emphysema subtyping.

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
