# OpenReview forum: "Deep Clustering Activation Maps for Emphysema Subtyping"
_MIDL.io/2021/Conference/Short — MIDL 2021 Poster_

### Official Review · Reviewer_aBSe · 2021-04-26

**Confidence:** 4
**Final Rating:** 3

**Summary:**

This work proposes the use of unsupervised deep clustering for the automatic segmentation of emphysema sub-typing from computed tomography scans.  Specifically, an iterative unsupervised process is proposed in which k-means clustering is first applied to deep features to obtain pseudo labels for then training the model in a classification task. The model training and the clustering algorithm iterate to converge to an unsupervised emphysema subtypes classification.


**Strengths:**

The paper is well written. The method is interesting and the results of the unsupervised approach are comparable with supervised ones. The activation maps are also interesting and add some interpretability to the algorithm.

**Weaknesses:**

Some details are missing and key steps of the method are difficult to understand. A Figure of the architecture and training would be of great help to understand but the space is probably limited.

How many clusters are used? I assume six for the emphysema subtypes.

“For each epoch, We“:This is a crucial typo which makes it difficult to understand the method.

“We generated dense clustering activation maps by applying g_W on dense features” They are applied to the feature maps (outputs of f(x)), not to dense features ? It is not fully clear how the CAMs are computed: “Dense clustering activation maps of four cases from two different clusters.” I would assume you do a rather standard CAM for each cluster, but this “two different clusters” is confusing.

The description of the baseline model isn’t clear: “The baseline method used a 3D-UNet without up-sampling layers as ...“

There are no repetitions that would show the variability of the method due to initialization and to population sampling (standard deviation, possibility to run statistical tests).


**Deanonymize Review:**

no

**Detailed Comments:**

Other comments:

I don’t know of many AIs that are “completely data-driven fashion.” Especially, your method uses manual annotations of the lung. Maybe rather use the term unsupervised which seems more appropriate?

“its parameters are denoted as g_W” g_W is the function/sub-model, the parameters are W

Sentence to fix: “Dense features are features after the”

I think the title is not really appropriate: The main point of the paper is the unsupervised segmentation I think. The CAM is an (interesting) extra interpretability step but should maybe not be in the title.


**Justification Of The Rating:**

Overall good short paper that I enjoyed reading, with interesting preliminary results that are worth sharing with the MIDL community.
I suggest to clarify important points prior to publication as mentioned above.


**Paper Type:**

methodological development

**Special Issue:**

no

---

### Meta-Review · Area_Chair_mhdF · 2021-05-09

**Recommendation:** Accept (Poster)
**Confidence:** 5

**Metareview:**

The high-quality review highlights the main positive aspects or a well-written paper with a good methodological contribution. I agree with their opinion and recommend acceptance. There are a few important suggestions that should be fixed for the final version.

---

### Decision · Program_Chairs · 2021-05-11

Accept (Poster)